# Physical Education Teachers’ Health Literacy: First Evidence from Lithuania

**DOI:** 10.3390/healthcare12131346

**Published:** 2024-07-05

**Authors:** Saulius Sukys, Laima Trinkuniene, Ilona Tilindiene

**Affiliations:** Department of Physical and Social Education, Lithuanian Sports University, Sporto 6, LT-44221 Kaunas, Lithuania; laima.trinkuniene@lsu.lt (L.T.); ilona.tilindiene@lsu.lt (I.T.)

**Keywords:** physical education teacher, health literacy, health behavior, HLS_19_-Q12

## Abstract

Background: Promotion of health literacy is an important goal in the context of promoting whole school health. Physical education teachers are of particular importance in achieving this goal. However, very limited empirical studies have addressed the health literacy of physical education teachers. This research aimed to test the structural validity and reliability of the HLS_19_-Q12, to measure health literacy among physical education teachers, and to evaluate associations of health literacy with health- and lifestyle-related indicators. Methods: We conducted a cross-sectional study of Lithuanian physical education teachers. The participants completed a self-administered online survey that collected information on socio-demographics and health literacy (HLS_19_-Q12 for general health literacy and the optional package HLS_19_-DIGI to measure digital health literacy) as well as health behavior. Results: A total of 332 participants completed the survey. The HLS_19_-Q12 demonstrated acceptable internal consistency (Cronbach’s α of 0.73 and McDonald’s ω of 0.75) and satisfactory structural validity (CFI = 0.924, TLI = 0.917, RMSEA = 0.081). Participants had an average score of 85.09 (SD = 17.23) when using the HLS_19_-Q12, with 19.3% and 48.8% displaying excellent and sufficient levels of health literacy, respectively. Regression analyses revealed that a higher level of health literacy was significantly associated with better health evaluation (β = 0.15, *p* < 0.01), but no significant association was found with other health behavior and lifestyle indicators. Conclusions: Overall, the results suggest that teachers’ general health literacy is relatively high. Our findings highlight the importance of conducting more in-depth studies to pursue the understanding of the whole school teachers’ health literacy.

## 1. Introduction

Health literacy is a concept that is both new and old [1], first being introduced in the 1970s [2]. As the idea of health literacy has many implications for health care, health education, and health promotion [1], it has become increasingly important for public health in the 21st century [3]. Overall health literacy is linked to “people’s knowledge, motivation and competences to access, understand, appraise, and apply health information in order to make judgments and take decisions in everyday life concerning healthcare, disease prevention and health promotion to maintain or improve quality of life during the life course” ([4] p. 3).

As research on health literacy increases, it is no coincidence that in the last decade, an increasing number of review studies were published. Systematic reviews of health literacy have shown associations between low health literacy and increased use of emergency care and hospitalizations, poorer medication-taking skills, poorer interpretation of health messages [5,6], having more problems communicating and navigating through the health care system [6], poorer health status [5], lower cancer screening rates within recommended guidelines [7,8], and higher mortality rates [5]. Low health literacy is related to higher hospital readmissions for coronary artery disease patients, higher anxiety levels, and lower social support [9]. The findings revealed that people experiencing heart failure have better self-care management if they have adequate health literacy [10]. Systematic reviews have also revealed that better health literacy is negatively related to smoking [11] and is positively related to quality of life [12] and the likelihood of being physically active [13,14].

Although health literacy is an important indicator of health outcomes, large-scale cross-cultural studies have revealed that a considerable proportion of adults have a low level of health literacy [15,16]. Moreover, although the promotion of health literacy is relevant to adults, it is even more important to focus on health literacy among children. In this context, the promotion of children’s health literacy at school is very important [17], especially as relatively few students have a high level of health literacy [18]. Educational intervention can play an important role in promoting health literacy [19], which goes on to have a significant impact on health behaviors and related outcomes [20,21]. Analysis of school-based health literacy interventions reveals the crucial role of teachers and the effects of their health literacy [22]. In the modern school health paradigm [23,24], teachers are not only subjects of school health but also valuable resources who should be supported in promoting healthy behaviors within schools.

However, the teaching profession is psychologically and physically demanding, with teachers encountering numerous stress factors in their work [25]. Experiencing prolonged stress can lead to burnout [26,27]. Compared to other professional groups, teachers more frequently suffer from mental and psychosomatic illnesses and nonspecific complaints [28]. Teachers experience higher levels of stress, burnout, anxiety, depression [29], exhaustion and fatigue, headaches, tension, listlessness, sleep and concentration disorders, inner restlessness, and increased irritability [30]. Therefore, behavior that strengthens the health of educators becomes crucial—both because it positively impacts the health of teachers themselves and because teachers who behave in such a way become examples of healthy behavior for their students. Previous studies have emphasized that teachers’ health literacy can have a significant influence on their own well-being; for example, studies show that teachers with high health literacy levels more frequently practice a healthy lifestyle [31] and exhibit greater occupational self-regulation [32]. Research has shown that health literacy is a strong predictor both of teachers’ health-promoting behaviors, such as healthy eating, physical activity, and stress management, and of a reduction in their risky habits [31]. A recent study in China showed that teachers who possess lower levels of health literacy tend to exhibit poorer health statuses, engage in more health-compromising behaviors, utilize health services more frequently, and incur higher healthcare costs [33]. Another study found that teachers with sufficient health literacy evaluate their own work as less intensive and of better quality than those with insufficient health literacy [34]. In this context, health literacy is an essential personal competence for enhancing teachers’ health-promoting and educational behaviors, because the health education of students is one of the priority directions of Lithuanian health policy [35]

Although teachers’ health literacy is associated with a wide range of health-related outcomes, research provides different data on teachers’ health literacy, which is not always sufficient. A study of Turkish teachers showed that 44.0% of them evaluated their own health literacy as very limited, 29.8% as limited, and 26.2% as being at an adequate level [36]. A study in Hong Kong revealed that 50.8% of school teachers had sufficient health literacy, 38.3% had problematic health literacy, and 10.9% had inadequate health literacy [34]. One of the most recent studies in China showed that more than half of teachers (56.9%) had inadequate or problematic health literacy [33]. The findings of the only study in Lithuania showed that 42.0% of teachers had insufficient health literacy [37]. 

Furthermore, few studies have also analyzed the determinants of teachers’ health literacy. Personal factors such as age, gender, work experience, and health have been identified as limited health literacy risk factors; teachers had low health literacy levels if they were men, older, or had longer work experience [33,36]. Additionally, chronic health conditions negatively predict health literacy [33].

To sum up, a very limited number of studies have focused on teachers’ health literacy. However, there is an even greater lack of research data on the health literacy of physical education teachers. Although the involvement of the entire school community in healthy lifestyle education is important [38], physical education teachers are of particular importance. The goal of physical education lessons and the content of these classes are directly related to schoolchildren’s health education. In Lithuania, for example, the goal and content of physical education as a school subject is regulated by the general physical education program, according to which its goals are to develop self-awareness and confidence in one’s abilities; to form and consolidate movement and physical activity skills; to provide knowledge and understanding for sustainable self-development; and to establish life-long attitudes and competences for the preservation and strengthening of health [39].

Scholars suggest that the health literacy, attitudes, and behaviors of physical education teachers influence the physical and mental health of students [40], who construct their own understanding through experiencing things and reflecting on those experiences [41]. The health behaviors of physical education teachers are shaped by their health knowledge, values, and social environment [42]. On the other hand, certain findings suggest that teachers may consider it challenging to teach health literacy if they themselves have low health literacy [43]. However, data on the level of physical education teachers’ health literacy is very limited. A study performed in Taiwan [44] found that the health literacy of health education and physical education teachers was satisfactory. But, it also revealed that teachers who exclusively taught physical education had a lower level of health literacy than those who taught health education. As there is a lack of data on the health literacy of physical education teachers, this study aims to fill this gap. 

Digital health literacy has recently also received attention when examining health literacy. The spread of health-related information, especially during the pandemic, in the digital space, especially on the internet and on social media, has prompted the attention of researchers [45,46]. The growing number of digital tools related to health and health-related information require not only the ability to use them [47] but also the ability to navigate the abundance of health-related information they generate or to which they provide access. Digital health literacy is defined as the ability to find, understand, and use health information from digital sources [48,49]. Digital technologies and access to health resources can be useful for improving teachers’ personal health literacy and teaching skills [42]. Findings also suggest that improving teachers’ digital health literacy could have a positive effect on their well-being and their interactions with children [50]. Although some studies have indicated that physical education teachers perceive themselves as having an intermediate level of digital teaching competence [51], no study has measured physical education teachers’ digital health literacy to date. 

It is important to state that many different measurements of health literacy have been developed [52]. Among them, the development and use of the 47-item European Health Literacy Questionnaire (HLS-EU-Q47) was important for encouraging health literacy research, as it was validated in different European countries [53], Taiwan [54], Japan [55], and six further Asian countries [56]. This questionnaire has also been used in health literacy studies in Lithuania [57,58]. However, this questionnaire is long and not always easy to use. It is worth mentioning that the Health Literacy Population Survey 2019–2021 (HLS_19_) (M-POHL), which included 17 countries in the WHO European Region [16], developed a shorter, 12-item version to measure general health literacy, known as HLS_19_-Q12 [16,59]. It is based on the same conceptual framework and definition of health literacy [4] as the HLS-EU-Q47 questionnaire. The good psychometric properties and usage feasibility of this new questionnaire [59] are a good basis for considering using it to measure the health literacy of physical education teachers. 

The scientific research discussed above suggests that physical education teachers can play an important role in promoting schoolchildren’s health. The effectiveness of their educational role is also linked to their health literacy. However, available empirical studies on the health literacy of physical education teachers are still missing. With this in mind, the research question of the study presented in this article is what is the health literacy of Lithuanian physical education teachers? To answer to this question, we aimed (a) to test the structural validity and reliability of the HLS_19_-Q12, (b), to measure the general and digital health literacy of physical education teachers, and (c) to evaluate the associations between health literacy and health- and lifestyle-related indicators. 

## 2. Materials and Methods

### 2.1. Study Design and Setting

For this study, we administered a cross-sectional online survey of physical education teachers from public schools in Lithuania. Following the Order of the Minister of Education, Science, and Sport of the Republic of Lithuania, only persons with higher education can work as physical education teachers in Lithuania [60]. General programs of physical education [39] regulate what knowledge and competences physical education teachers must develop in students. Teachers of physical education (as well as other educational subjects) can have qualifications in the following categories: teacher, senior teacher, teacher methodologist, and expert teachers These qualification categories are distinguished not only by the teaching experience the possessor has but also by the reach of their organizational and good pedagogical experience. For example, a senior teacher is able to spread his/her good pedagogical experience in a school, a methodologist in a school and in a region, and an expert nationwide.

### 2.2. Study Participants and Procedures

In this study, the research participants were physical education teachers from Lithuania. The required sample size for the study was determined based on the total number of physical education teachers in the country’s public schools. Choosing a 95% confidence level and ±5% confidence interval, the minimum sample size was 310 teachers. In assessing the required number of subjects, we considered not only the size of the population but also our intention to test the structural validity of the health literacy scale. Some guidelines recommend at least 10 participants per scale item [61]. Health literacy was measured using a 12-item scale; a minimum sample size of 120 might therefore have been adequate. Since exploratory and confirmatory factor analysis must be performed with different subjects [62], the estimated number of subjects of 310 was sufficient.

Before initiating the study, the research protocol was approved by the Research Ethics Committee of the Lithuanian Sports University. The data were collected via an online survey between November 2023 and January 2024. The survey was implemented through the Google Form platform. Invitations to participate in the study were distributed with the assistance of the Lithuanian Association of Physical Education Teachers. This ensured that all the physical education teachers received information about the study. Prior to completing the survey, the participants were introduced to the study aims and study measures. It was explained to them how anonymity and confidentiality would be ensured and approximately how long it would take to complete the questionnaire. The study participants were informed that they could refuse to participate in the study or withdraw from it at any time by closing their browser. After providing informed consent digitally, the participants were directed to the questionnaire. The survey form was restricted to accepting only one response per IP address.

After obtaining permission from the HLS_19_ Consortium, translations of the health literacy measurement questions (HLS_19_-Q12 version) and one optional package—Digital Health Literacy (HL-DIGI, HL-DIGI-INT, and HL_DIGI-DD)—into the Lithuanian were made according to guidance provided by the HLS_19_ Consortium [16], as well as other [63] recommendations. More specifically, two translations from English into Lithuanian were made by professional translators, and following a comparison of the two translations, a decision on the most appropriate was reached via the consensus of experts in health education and the research team. This was followed by a back-translation into English by another two professional translators. The final translation was reviewed by the health education scientist and discussed with the translators to ensure the content had a comparable meaning and to maintain item clarity and semantic equivalence. Finally, a pilot study involving 35 physical education teachers was conducted. 

### 2.3. Study Measures

#### 2.3.1. General Health Literacy

In this study, we used the HLS_19_-Q12 instrument, which is a newly developed 12-item short-form questionnaire of the HLS_19_-Q47 for measuring comprehensive, general health literacy in general adult populations [16,59]. This instrument was introduced with the following statement “It is not always easy to get understandable, reliable, and useful information on health-related topics. With the following questions we would like to find out which tasks related to handling health information are more or less easy or difficult. On a scale from very easy to very difficult, how easy would you say it is…”. Participants were asked to choose from one of four response categories: 4 “Very easy”; 3 “Easy”; 2 “Difficult”; and 1 “Very difficult”. The HLS_19_-Q12 showed adequate internal consistency, with an average Cronbach’s alpha of 0.78 across 17 countries [16,59].

By analyzing the data, we first summarized the distributions of the responses to individual items in terms of the percentages of those who responded “very difficult” or “difficult” to each item (answers were dichotomized). Second, we calculated the Average Percentage Response Patterns (APRPs) as the average of the percentages of how often each category was selected for all items by each respondent. Third, we calculated the overall score of the HLS_19_-Q12 as the percentage (ranging from 0 to 100) of items with valid responses that were answered “very easy” or “easy”, provided that at least 80% of the items contained valid responses [59]. Higher scores indicate a higher level of general health literacy. Finally, we calculated the levels of health literacy following the recommended procedures (for the detailed procedures, see Pelican et al. [59]). Four levels of health literacy were identified: excellent, sufficient, problematic, and inadequate. 

#### 2.3.2. Health Literacy Optional Package

In the M-POHL project [16], the optional health literacy packages included the Digital Health Literacy (HLS_19_-DIGI), the Navigation Health Literacy (HLS_19_-NAV), and the Vaccination Health Literacy (HLS_19_-VAC) packages. In our study, we used the Digital Health Literacy package. More specifically, we used the HLS_19_-DIGI 8-item questionnaire, which was developed by a working group of the HLS_19_ [16] to measure digital health literacy in the adult population. This questionnaire was introduced with the following statement: “When you search online for information on health, how easy or difficult is it for you”. In addition, this 8-item questionnaire was accompanied by an additional 2 items to measure interaction with digital devices that began with the following statement: “When typing a health-related message on a digital device how easy or difficult is it for you to”. Response categories for both scales ranged from 1 “Very difficult” to 4 “Very easy”. 

We used an additional 6-item questionnaire developed by the same working group [16] to measure the frequency of use of digital resources. Research participants had to reply to the question “In a typical week, how many days do you use the following digital resources for getting health related information?” by choosing response categories from a choice of “Not relevant for me”, “Less than once per week”, “1–3 days per week”, “4–6 days per week”, “Once a day”, or “More than once per day”. The list of resources included websites, social media, digital devices, mobile health apps, eHealth, and other digital resources).

By analyzing the data, we first summarized the distributions of the responses to individual items on the 8-item scale according to the percentages of those who responded “very difficult” or “difficult” per item. Second, we calculated the APRP in the same way as for general health literacy. Third, we calculated the overall HLS_19_-DIGI score ranging from 0 to 100 [59]. Higher score values indicate a higher level of digital health literacy. For interactions with digital devices, we calculated the APRP. 

For the use of digital devices for health, we calculated the percentage distribution of the responses to all 6 items. Also, a mean score (ranging from 1 “Not relevant or less than once per week” to 5 “More than once per day”) was calculated to measure the relative frequency of using health-related digital resources

#### 2.3.3. Health Behavior and Lifestyles

Health behavior and lifestyle variables included body mass index (BMI), physical activity, smoking behavior, alcohol consumption, and health status. 

BMI was measured by asking “How tall are you without shoes (in cm)?” and “How much do you weigh without clothes and shoes (in kg)?”. BMI was calculated as weight (kg) divided by height (m) squared. BMI was classified into normal BMI < 25.0, overweight BMI 25–29.9, and obesity BMI ≥ 30 [64].

Physical activity was measured as in the M-POHL project [16] by asking “How many days per week are you physically active (e.g., doing sports or other leisure time activities, at work, doing household or gardening chords or commuting from one place to another) for a total of 30 min or longer, evoking at least a small increase in breathing or heart rate?” Responses were from −1 “Never” to 7 “7 days”. By analyzing the data, we grouped the answers into four categories: never, occasional use (one day per week or less), light use (two or three days), and heavy use (from four to seven days).

For the smoking behavior measure, participants were asked to answer the questions “Have you smoked tobacco products during the last 12 months and during the last month?” and “Have you smoked electronic cigarettes during the last 12 months and during the last month?”. The response options were “No”, “Yes, at least once”, “Yes, occasionally”, and “Yes, every day” [65]. 

For the alcohol consumption measure, participants were asked to answer the questions “Have you consumed any alcoholic beverages during the last 12 months?” and “Have you consumed any alcoholic beverages during the last month?” by choosing from the response options “No”, “Yes, at least once”, “Yes, occasionally”, and “Yes, every day” [65]. 

Health status was measured by asking “How is your health in general?” with five answer options: “Very good”, “Good”, “Fair (i.e., neither good nor bad)”, “Bad”, and “Very bad”. 

### 2.4. Statistical Analysis

For the data analyses, we used the IBM SPSS statistics 29.0 (IBM, Armonk, NY, USA) and JASP 0.18.3 (University of Amsterdam, Amsterdam, The Netherlands) software. Before the main analyses, we tested the structure of the HLS_19_-Q12 instrument. First, we conducted exploratory factor analysis (EFA) followed by confirmatory factor analyses (CFA) using JASP. The data were randomly split into two samples [62]. The Kaiser–Meyer–Olkin (KMO) test was used to check whether the data were suitable for factor analyses. When conducting CFA, we selected the most commonly used fit indices [66]: the Root Mean Square Error of Approximation (RMSEA) (≤0.08), Comparative Fit Index (CFI), and Tucker–Lewis Index (TLI) (≥0.90) [67,68]. However, the model can be evaluated as possible with RMSEA values of no more than 0.10 [69]. Considering the recent criticism of the use of Cronbach’s alpha (α) [70,71], we assessed the reliability of the HLS_19_-Q12 using Cronbach’s alpha and McDonald’s omega (ω). For descriptive statistics, we calculated the distribution of the research participants’ characteristics, health literacy, digital health literacy and health behavior, and lifestyle variables. We conducted correlational analyses to measure correlations between the study variables. Using the chi-square test, we undertook comparisons of the study variables by socio-demographics. We conducted a multinomial regression analysis, first to measure predictors of teacher health literacy and second to measure health literacy as a predictor of health behavior and lifestyle. 

Regarding the STROBE statement, we followed the checklist guidelines in organizing this paper (Appendix A).

## 3. Results

### 3.1. Sociodemographic Characteristics of Respondents

In total, 356 participants participated in this survey. Considering that 24 (6.7%) physical education teachers had incomplete data on their physical literacy, the data of 322 participants were analyzed. Considering that the representative sample should have been of at least 310 subjects, the final sample of the study was sufficient. Almost two-thirds of them were female (61.4%), and the mean age was 51.0 years (SD = 9.96). Table 1 shows the distribution of other socio-demographic characteristics, such as age group, experience of working as a physical education teacher, professional qualifications, and classes taught by teachers.

### 3.2. General Health Literacy

#### 3.2.1. Distributions of the HLS_19_-Q12 Items

First, the distributions of the responses to the individual items of the HLS_19_-Q12 were summarized in two ways: by the percentages of those who responded with “very difficult” or “difficult” per item and by the average percentage of how often each response category was selected for all items by each participant. The overall percentage of respondents choosing “very difficult” or “difficult” varied between 3.9% and 37.8% for the HLS_19_-Q12 items (Table 2), with the items “to find information on healthy lifestyles such as physical exercise, healthy food or nutrition” and “to judge how your housing conditions may affect your health and well-being” being the easiest, and the item “to judge the advantages and disadvantages of different treatment options” being the most difficult. 

The study results showed that the physical education teachers most often responded to the HLS_19_-Q12 items with “easy” (63%) and “very easy” (22%). Another 14% of the participants answered “difficult”, and only 1% answered “very difficult”.

#### 3.2.2. Factorial Structure and Reliability of the HLS_19_-Q12

Before further data analysis, we checked the structure of the HLS_19_-Q12 by conducting EFA. Our analysis showed that the sample adequacy was met (KMO = 0.911), and there was a significant correlation between the variables (χ^2^ (66) = 1805.234, *p* < 0.001). One factor emerged that explained 47.89% of the variance, with the factor loadings ranging from 0.66 to 0.73. The scree plot of the HLS_19_-Q12 is presented in Figure 1.

Next, we conducted confirmatory factor analysis to assess the on-factor structure of the HLS_19_-Q12. This study revealed that a single-factor model of the 12-item health literacy questionnaire demonstrated an acceptable data fit: CFI = 0.924, TLI = 0.917, RMSEA = 0.081 (95% CI = 0.048, 0.096). Figure 2 shows the model structure. 

To assess the reliability of the HLS_19_-Q12, we assessed Cronbach’s α and McDonald’s ω. A Cronbach’s α of 0.73 and McDonald’s ω of 0.75 showed that the Lithuanian version of this questionnaire had an acceptable level of reliability.

#### 3.2.3. Distributions of the Score Values and Levels

Next, we calculated the standardized general health literacy scores, with scores ranging from 0 to 100. The average total health literacy score was 85.09 (SD = 17.23), and the 75% quantile was close to or equal to the maximum value of 100. Comparison by socio-demographic indicators did not reveal any statistically significant differences. 

Regarding levels of health literacy, 19.3% and 48.8% of the physical education teachers had excellent and sufficient levels of health literacy, respectively. On the other hand, 24.10% and 7.8% of the respondents had problematic and inadequate levels of health literacy, respectively. A comparison of health literacy levels by socio-demographic indicators revealed that the teachers’ health literacy significantly differed depending on their teaching experience (ꭓ^2^ (9, *n* = 332) = 19.69, *p* < 0.05), i.e., teachers with the longest working experience mostly had inadequate levels of health literacy and, correspondingly, the least excellent level of health literacy.

### 3.3. Digital Health Literacy

The digital health literacy package encompasses skills such as searching for, accessing, comprehending, evaluating, verifying, and utilizing online health information (assessed by eight specific items, referred to as HL-DIGI). The percentages of combined “difficult” and “very difficult” answers for the items of the HL-DIGI scale ranged from 13.9% for the easiest item “to understand the information” to 36.1% for the most difficult item “to judge whether the information is offered with commercial interests” (Table 3). The average percentage of how often each response category was selected for all items by each participant showed that the teachers most often responded to the HLS-DIGI items with “easy” (64%) and “difficult” (24%). Just 10% answered with “very easy”, and only 2% answered with “very difficult”. Cronbach’s α value was 0.73, and McDonald’s ω was 0.87.

We calculated the HL-DIGI scores ranging from 0 to 100. The mean score was 74.69 (SD = 30.94), and the 75% quantile was close to or equal to the maximum value of 100. Comparison of the HL-DIGI scores by socio-demographic indicators did not reveal any statistically significant differences. 

We assessed the teachers’ estimations of the ease or difficulty with which they interacted with digital media using a two-item set concerning the topic (HL-DIGI-INT). Our results revealed, that on average, the item “to clearly formulate your written message when communicating with a health provider” was considered easier (on average, 17.5% responded “difficult” or “very difficult”) than the item “to express your opinion, thoughts or feelings, ask a question in writing on social media including online forums” (on average, 26.2% responded “difficult” or “very difficult”). The study results showed statistically significant differences comparing the different age groups in terms of the teachers’ abilities to express their opinion, thoughts, or feelings or to ask a question in writing on social media (ꭓ^2^ (12, *n* = 320) = 31.04, *p* < 0.01) as well as to clearly formulate a written message when communicating with a health provider’s media (ꭓ^2^ (8, *n* = 331) = 23.02, *p* < 0.01). 

We also measured the frequency with which the physical education teachers used different digital sources and resources for promoting their health (Table 4). We found that the teachers most frequently used digital devices related to health or health care, followed by health apps on cell phones. Digital interaction with the health system was the least frequently used option. The mean score (ranging from 1 “Not relevant or less than once per week” to 5 “More than once per day”) was 1.84 (*SD* = 0.86). This study revealed that teachers of different teaching experience used digital devices related to health or health care (ꭓ^2^ (15, *n* = 332) = 35.01, *p* < 0.01) and health apps (ꭓ^2^ (15, *n* = 332) = 37.16, *p* < 0.001) differently. Specifically, digital devices and apps were rarely used by the teachers with the greatest teaching experience and were most often used by the teachers with no more than 10 years of teaching experience.

### 3.4. Determinants of General and Digital Health Literacy

We investigated socio-demographic factors as possible determinants of general health literacy and digital health literacy. First, we conducted a correlational analysis. We found no statistically significant relationships between general health literacy and socio-demographics. We did find that general health literacy was positively related to digital health literacy (r = 0.61, *p* < 0.01). Digital health literacy was significantly negatively related to teaching experience (r = −0.13, *p* < 0.05). Additionally, digital health literacy was positively correlated with the use of digital resources (r = 0.11, *p* < 0.05). 

Next, the multiple linear regression analysis revealed that the model assessing the effect of socio-demographic variables on general health literacy was not significant (*F* (5, 327) = 1.21, *p* > 0.05; R^2^ = 0.02). However, focusing on different predictors’ analyses revealed that age was positively (β = 0.19, *p* < 0.05) associated with general health literacy (Table 5). This indicates that being older predicts higher health literacy. We also investigated socio-demographics and use of digital resources as possible determinants of digital health literacy (Table 5). Any of these determinants were predictors of digital health literacy. 

### 3.5. Health Behavior and Lifestyles

Only 2.0% of the physical education teachers were categorized as never performing physical activity, while more than two thirds were categorized as heavy performers (Table 6). Regarding smoking behavior, most of the responders never used tobacco during the last 30 days as well as during the last year. On average, only 8% of the teachers used tobacco every day (Table 6). Even fewer teachers used electronic cigarettes. The survey showed that half of the teachers had occasionally used alcohol during the last 30 days, and even more (61.4%) had used alcohol during the last year. By assessing self-perceived health, we found that 60.2% reported their personal health as “good” and 12.7% as “very good”. A quarter perceived their personal health as fair. For BMI, 42.2% were classified as overweight, and 10.2% were classified as obese. 

We studied the correlation of health behavior and lifestyle indicators with health literacy. As Table 7 shows, only physical activity and self-perceived health were significantly correlated with general health literacy. We assessed any significant correlations of health behavior and lifestyle indicators with digital health literacy. 

### 3.6. Effects of Health Literacy on Health Behavior and Lifestyle

We conducted multiple linear regression analysis, including the five socio-demographic indicators and general health literacy as predictors and health behavior and lifestyle indicators as dependent variables (Table 8). There was no evidence of multicollinearity in any model (means of VIF > 4). The regression analysis revealed that after controlling for socio-demographic factors, general health literacy was significantly positively associated with self-perceived health, and the model explained 8% of the variance (*F* (6, 326) = 4.31, *p* < 0.001). The regression model explained 7% of the variation in physical activity (*F* (6, 326) = 4.75 *p* < 0.001). However, general health literacy was not a significant predictor. Regression models were significant for e-cigarette use per/over the last year (*F* (6, 326) = 4.51, *p* < 0.001) and over the last 30 days (*F* (6, 326) = 6.28, *p* < 0.001) and for alcohol use during the last 30 days (*F* (6, 326) = 2.15, *p* < 0.05). However, for all these health behavior indicators, general health literacy was not a significant predictor. Finally, the regression model for BMI was also significant (*F* (6, 326) = 12.84, *p* < 0.001), explaining 19% of the variance. A significant effect of general health literacy on BMI was not revealed.

No correlation was found between the physical education teachers’ digital health literacy, health behavior, and lifestyle indicators. Therefore, we did not evaluate the effect of digital health literacy on health behavior and lifestyle.

## 4. Discussion

The authors of this study attempted to determine the health literacy of Lithuanian physical education teachers. The main research instrument used in this study was the HLS_19_-Q12, which was used to assess teachers’ general health literacy. As we found no data on this research instrument’s use for studying health literacy in Lithuania, our first goal was to check its structural validity and reliability.

Previous studies demonstrated the unidimensional structure of this research instrument [16,59,72,73,74]. The results from the EFA and CFA confirmed that the Lithuanian version of this questionnaire had a one-dimensional factor structure. The research revealed that the scale was sufficiently reliable. More specifically, for dichotomous items, the Cronbach score was above the recommended acceptable level [75], as was McDonald’s omega. In the M-POHL [16,59], the value of Cronbach’s alpha for dichotomous items ranged from 0.67 to 0.87. However, it was noted that when using a non-dichotomous items scale (four-point rating scale), the reliability values for Cronbach’s alpha were higher [59]. After having checked the latter proposition, we also found that for the four-point scale, Cronbach’s alpha was higher (0.88). We also want to highlight that we calculated both Cronbach’s alpha and McDonald’s omega. Recently, it has been discussed, with practical examples, that it is better to use omega when checking the reliability of a scale [70,71]. However, if studies use an already developed questionnaire whose reliability has been determined using the alpha of previous studies, we recommend using alpha along with omega. This would allow researchers to compare the reliability estimates with previous studies and, at the same time, notice differences in the calculation of reliability when performed by different methods. In addition, it is not recommended to choose omega because the assumptions of alpha are not met [70].

We evaluated each statement by calculating the percentage of participants who chose the answer options “very difficult” or “difficult”. The study results showed that the responses ranged between 3.9% and 37.6%. Although the comparison of the results in our study with study results from 17 European countries [16] is inexact due to the specific group of subjects, some similar trends are worth mentioning. For example, in the European health literacy survey as well as in our research, the most difficult item was “to judge the advantages and disadvantages of different treatment options”. In our research, the item “to find information on healthy lifestyles such as physical exercise, healthy food or nutrition” was the second easiest, which is also consistent with the results from the aforementioned project [16,59]. Interestingly, the item “to judge how your housing conditions may affect your health and well-being” was the easiest for physical education teachers, while this item was not among the easiest among 17 European countries. 

Although scale validity and reliability are important, the second aim of our study was no less or even more important for determining physical education teachers’ health literacy. Our study revealed that the general health literacy score of the physical education teachers was 85.09, which indicates a sufficiently high level of literacy. Compared to the above-mentioned health literacy studies in 17 European countries [16], the general health literacy score of our study’s physical education teachers is higher.

When evaluating teachers’ health literacy, we considered how the subjects were distributed according to health literacy levels to be important. Our study revealed that almost 70% of the physical education teachers had excellent or sufficient levels of health literacy. As the amount of previous research involving physical education teachers is very limited, the possibilities for comparison were similarly limited. However, when compared to the results in the Taiwan study of physical education teachers [44], similar trends can be observed, as a satisfactory level of health literacy was determined there. However, a comparison of the health literacy data of physical education teachers with teachers of other subjects studied in other countries [33,34,36] revealed the lower health literacy level of the latter. Similar differences were observed when we compared our study data with the data in a study involving teachers of different educational subjects in Lithuania [37]. The data from this latter-mentioned study of teachers almost coincided with the data from the population study on the health literacy of Lithuanian adults [76]. Consequently, if the health literacy of teachers generally reflects similar trends in the health literacy of the adult population, the health literacy of physical education teachers is of a higher level. We can assume that these differences are determined by the physical education teachers’ competencies related to health education, which are necessary for them to realize the goal of physical education as an academic subject.

By analyzing the determinants of general health literacy, we found a significantly positive association between health literacy and age. Previous studies showed that health literacy declines as age increases [16,77,78,79]. Similar results were found in studies involving schoolteachers [33,36]. On the other hand, no association between teachers’ ages and health literacy was found in a previous study of physical education teachers [44]. Therefore, the data we obtained suggest that health literacy does not always decrease with increasing age. These associations also depend on various other factors, such as the presence of health problems and the number of contacts with physicians over the preceding year [78]. The data we obtained should encourage researchers to consider the nature of the subjects’ work when examining the relationship between age and health literacy. We would like to highlight another aspect when discussing age. More specifically, when evaluating the impact of age and comparing data from different studies, it is important to consider the specific ages of the teachers being studied. The average age of the teachers who participated in our study was 51 years, indicating that they were relatively mature. This is not surprising; rather, it reflects the general situation of teachers in Lithuania. Overall, the average age of all teachers in the country in 2023–2024 is 50.74 years [80]. By the way, 8.85% of teachers are of retirement age, and this percentage has been increasing over the past ten years (6.5% in 2015) [80]. Research has also revealed that teachers under the age of 50 are 1.35 times more likely than older teachers to express a desire for additional information about strengthening students’ health [81]. Therefore, it is important to critically evaluate when comparing data on teachers’ health literacy (and other health-related indicators) from different studies by age.

We evaluated teachers’ ability to search for, access, understand, appraise, validate, and apply online health information that refers to digital health literacy in this study along with their general health literacy. We found that the mean score for digital health literacy was lower than that for general health literacy. For physical education teachers, it was difficult to determine whether the online information was offered with commercial interests, whether it was reliable, and whether it applied to them. However, the lack of studies with physical education teachers limits comparison possibilities. By comparing our data with the data derived from a survey of the adult population in European countries [16], we noticed that the digital health literacy of physical education teachers is of a higher level. Some trends are similar; the most difficult task when searching for health-related information online is to assess its reliability, personal applicability, and the commercial nature of the information.

Our survey revealed that of the different digital sources and resources for promoting their health, the physical education teachers most often used a digital device related to health or health care and various health apps on mobile phone. These results are not surprising, because an exponential increase in the number of various health apps was predicted a decade ago [82]. Despite extensive commercialization in this field, research evidence has shown that the use of health apps is related to a higher level of education, better personal health evaluations, increased physical activity, a higher level of health literacy, and the intention to engage in health-promoting activities [83]. An assessment of systematic reviews also confirmed that the use of various activity trackers (fitness trackers, activity-tracking smartwatches, and pedometers) could be effective in increasing children’s and adults’ physical activity [84]. The purpose of physical education lessons is to develop healthy lifestyle skills, including physical activity. Moreover, physical education teachers rate the integration of health apps in physical education positively [85]. Other research revealed that the use of various digital technologies can be useful for achieving physical education goals, as they increase schoolchildren’s engagement in physical education [86] and their motivation and improve sport-specific motor capabilities and skills [87]. Therefore, the results we obtained are not surprising.

Previous studies revealed that various socio-demographic aspects (such as gender, age, education, social deprivation, and the use of digital resources) are significant determinants of adult digital health literacy [16]. We also analyzed determinants of digital health literacy, but the study results did not reveal any relationship between socio-demographic factors and the use of digital resources. It is important to mention that we did not compare the data by education level, as all teachers in the country, including physical education teachers, are required to have a higher education degree [60]. 

The final aim of our study was to evaluate the associations of health literacy with health- and lifestyle-related indicators. Previous studies of adults revealed that health literacy is a significant positive predictor of health status [16] and physical activity [13,14,16], while it is a negative predictor for smoking [11,16] and alcohol use [16]. Similarly, studies of teachers revealed that health literacy is positively related to their physical activity and better personal health evaluation [31,33]. Our study revealed that the general health literacy of the physical education teachers was positively related only to a better assessment of personal health, which repeats the results obtained in previous studies of both the adult population and, specifically, of teachers. However, no significant relationships were found between health literacy and other indicators of health behavior and lifestyle. These data allow us to reason that the revealed relationships among the adult population or even among teachers are not repeated in the context of physical education teachers. To explain this, the lifestyle indicators of the studied teachers should be considered.

Our study revealed that the majority of the physical education teachers were physically active, which repeats similar results obtained in other studies [88]. Such results are not surprising, because the work of a physical education teacher requires being physically active, with a high work intensity and a high level of fitness being typical of their profession [89]. According to the latest research on the lifestyle of Lithuanian adults, only 28.3% of adults engage in vigorous physical activity for at least 30 min five days or more per week [90]. We found that less than 10% of the physical education teachers smoked daily. Regarding smoking and alcohol consumption, previous studies have provided evidence that smoking is generally less common among teachers than in the general adult population [30,91] or professionals working in the health, social, or educational fields [92]. It is worth mentioning that among the adult population of Lithuania [90], there are twice as many daily smokers (16.3%) than there were among the physical education teachers. The same differences were observed when evaluating alcohol consumption [91]. These data suggesting physical education teachers’ good physical activity levels and few bad habits ultimately lead to the fact that there was no statistically significant relationship between general health literacy and lifestyle indicators.

No association was found between general health literacy and BMI. A comparative study of adults from 17 European countries [16] revealed a significant relationship between general health literacy and BMI in only two countries. In addition, some studies from non-European countries did not find a significant relationship between health literacy and overweight status [93]. Although we lack studies identifying physical education teachers’ BMI, comparison with teachers’ data revealed interesting similarities and differences. In our study, 10% of the physical education teachers were obese. In other studies, the percentage of obese teachers varies from 12% [30] to 15.9% [92]. However, when we compared those subjects whose BMIs were ≥25 kg/m^2^, the results were not similar. In our study, half the teachers belonged to this group, as compared to another study’s value of 24% [91]. The results we obtained were somewhat unexpected, especially considering that physical education teachers are physically active. On the other hand, further comment on BMI as a health indicator is needed. Although BMI is widely used as a measure of weight status, it also receives considerable criticism [94,95]. One limitation of BMI is that it is not a good indicator of body fat mass [95]. Thus, many factors such as gender, age, obesity in families, lifestyle, and genetic factors must be considered when interpreting BMI data [95]. As we mentioned earlier, the specific work carried out by a physical education teacher requires a high work intensity and a high level of fitness [89]. It is likely that this target group will tend to have more muscle mass, which may affect their BMI categorization. Thus, when comparing BMI data, it is important to take into account the studied group and its specifics. Therefore, in further studies, when studying the lifestyle of physical education teachers, it would be appropriate to investigate specifically the links between their physical activity and BMI in more detail.

In summary, the strength of our study is that it is the first study in Lithuania and one of the few generally to assess the health literacy of physical education teachers. This study revealed important data about physical education teachers’ health literacy. It also confirmed that the Lithuanian version of the HLS_19_-Q12 possesses sufficient structural validity and reliability. We also examined the effect of health literacy on health behavior and lifestyle. Although more relationships could have been expected, only a significant positive relationship with personal health evaluation was found. Considering the specific nature of the participants of this study, we have several suggestions for further research. Interactivity, emotionality, time pressure, and lack of freedom at work are aspects of the work of teachers in general and of physical education teachers in particular, and they can easily cause stress [25], burnout, anxiety, and depression [29]. Psychological tension and consequent psychological problems are also related to cardiovascular diseases or risk factors of cardiovascular diseases [96]. Therefore, in continuing this research, it would be relevant to examine how teachers’ health literacy is related to various psychosocial work strains. It would also be useful to conduct research on teachers of different subjects. This would allow us to compare how much data and relationships with other indicators of physical education teachers’ health literacy differ from the data of teachers of other subjects. Moreover, interdisciplinarity coherence between physical education and other subjects is recommended in Lithuania [97], such as biology, whose educational content can also cover many healthy lifestyle topics. Therefore, health literacy data would reveal whether the health literacy of teachers who teach subjects whose educational content is more related to promoting a healthy lifestyle differs

Digital health literacy was used as a supplementary variable in the present study. In order to promote students’ physical activity, there are ever greater opportunities for various digital technologies to be used in the work of physical education teachers. Therefore, in further research, it would be important to analyze how teachers’ digital health literacy is related to the use and exploitation of digital technologies for health promotion.

Our study is not without limitations. Although the reliability of the HLS_19_-Q12 was very good, we did not examine test–retest reliability, nor was test–retest reliability examined in the European context [16]. Some recent studies in China found that this measure possessed moderate test–retest reliability [74], which indicates that it is important to address this in future research. We did not create and validate a new research instrument, and we only assessed the structural validity of the HLS_19_-Q12 and found that it is a unidimensional instrument sufficient to measure general health literacy. We did not assess convergent validity. Other limitations of the study relate to both the measurement of lifestyle indicators and the participants of the research. We assessed physical activity and divided the participants into physical activity groups in the same way as the project featuring 17 European countries [16]. Future research should consider more rigorous measures of physical activity. Among the socio-demographic data, we collected information on teachers’ qualification status. However, this category is not directly related to teachers’ competencies in the field of health education. Another additional important indicator would be whether the school participates in health education projects, such as the health promotion school network. Finally, this study was cross-sectional, and the relationship between teachers’ health literacy and health behavior should not be interpreted as causality.

## 5. Conclusions

In this study, we provided important insights into physical education teachers’ health literacy. The results suggest a relatively high level of teachers’ general health literacy. Although the use of various digital and smart technologies related to health and the monitoring of its indicators are common enough among physical education teachers, it is more difficult for them to evaluate online health information. In contrast to previous studies of the adult population, our study found few relationships between health literacy, health, and lifestyle. Such data might be a result of the prevalence of relatively good health behavior. Our findings highlight the importance of undertaking further in-depth studies to better understand schoolteachers’ health literacy generally. This would permit not only a comparison of the health literacy of teachers of different subjects but also foster a better understanding of the extent to which teachers’ health literacy can be used to promote a healthy lifestyle among students in different subjects as well as assisting in creating and implementing general health-promoting policies at school.

## Figures and Tables

**Figure 1 healthcare-12-01346-f001:**
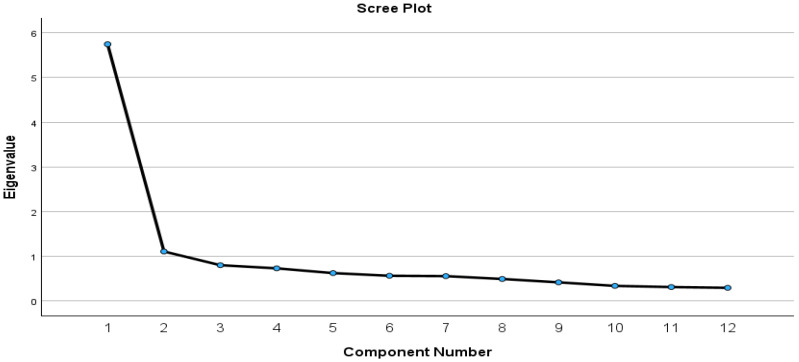
The scree plot of the HLS_19_-Q12.

**Figure 2 healthcare-12-01346-f002:**
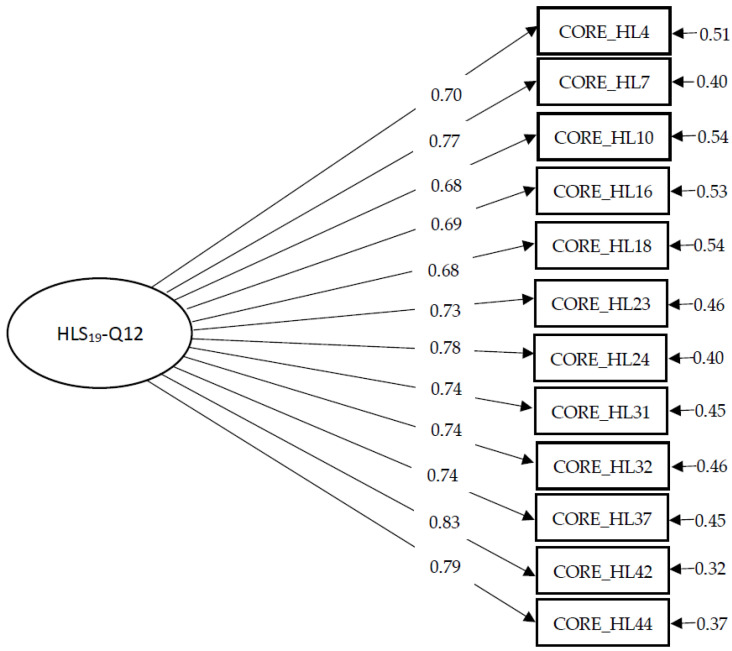
Unidimensional HLS_19_-Q12 model from confirmatory factor analysis.

**Table 1 healthcare-12-01346-t001:** Sample characteristics (*n* = 332).

Variable	Categories	*n*	%/M (SD)
Age		322	51.00 (9.96)
Age groups	26–35	26	7.8
	36–45	71	21.4
	46–55	112	33.7
	56–65	113	34.1
	65+	10	3.0
Gender	Female	204	61.4
	Male	128	38.6
Years in role	≥10	46	13.9
	11–20	57	17.1
	21–30	100	30.1
	30+	129	38.9
Teacher qualification	Teacher	29	8.7
	Senior teacher	90	27.1
	Teacher methodologist	181	54.5
	Expert	32	9.7
Classes taught by teacher	Only primary	16	4.8
	Basic	98	29.5
	Secondary	218	65.7

Note: *n*: number of participants; %: percentage; M: mean; SD: standard deviation.

**Table 2 healthcare-12-01346-t002:** Descriptive results of the HLS_19_-Q12 items.

Item Number	Question: “On a Scale from Very Easy to Very Difficult, How Easy Would You Say It Is…”	Mean (SD)	Percentages of “Very Difficult” or “Difficult” Responses (%)
1	to find out where to get professional help when you are ill?	1.91 (0.28)	8.7
2	to understand information about what to do in a medical emergency?	1.91 (0.28)	8.7
3	to judge the advantages and disadvantages of different treatment options?	1.62 (0.49)	37.8
4	to act on advice from your doctor or pharmacist?	1.93 (0.26)	7.2
5	to find information on how to handle mental health problems?	1.71 (0.45)	28.9
6	to understand information about recommended health screenings or examinations?	1.83 (0.37)	16.6
7	to judge if information on unhealthy habits, such as smoking, low physical activity or drinking too much alcohol, are reliable?	1.91 (0.28)	8.7
8	to decide how you can protect yourself from illness using information from the mass media?	1.77 (0.42)	23.2
9	to find information on healthy lifestyles such as physical exercise, healthy food or nutrition?	1.94 (0.23)	5.7
10	to understand advice concerning your health from family or friends?	1.91 (0.29)	9.3
11	to judge how your housing conditions may affect your health and well-being?	1.96 (0.19)	3.9
12	to make decisions to improve your health and well-being?	1.81 (0.40)	19.3

**Table 3 healthcare-12-01346-t003:** Descriptive results for the HLS-DIGI items.

Item Number	Question: “When You Search Online for Information on Health, How Easy or Difficult Is It for You”	Mean (SD)	Percentages of “Very Difficult” or “Difficult” Responses (%)
1	to judge whether the information is reliable?	1.67 (0.47)	32.5
2	to judge whether the information is offered with commercial interest?	1.63 (0.48)	36.1
3	to understand the information?	1.86 (0.35)	13.9
4	to use the information to help solve a health problem?	1.76 (0.43)	23.8
5	to judge whether the information is applicable to you?	1.69 (0.46)	30.4
6	to find the exact information you are searching for?	1.70 (0.46)	29.5
7	to visit different websites to check whether they provide similar information about a topic?	1.81 (0.39)	19.0
8	to use the proper words or search query to find the information you are looking for?	1.83 (0.38)	16.9

**Table 4 healthcare-12-01346-t004:** Percentage distributions of the responses to use digital resources.

Type of Digital Resources	Not Relevant/ DK/Refusal	Less than Once per Week	1–3 Days per Week	4–6 Days per Week	Once a Day	More than Once per Days
Websites	4.2	55.1	21.4	8.1	5.1	6.0
Social media including online forums	7.5	56.9	17.5	8.7	3.6	5.7
A digital device related to health or health care	4.2	31.6	17.8	17.8	14.8	13.9
Health app on your mobile phone	6.6	40.1	19.6	10.5	15.1	8.1
Digital interaction with your health system	13.9	60.8	16.0	5.4	3.3	0.6
Other	25.4	50.5	14.8	3.6	3.6	2.1

Note. DK = do not know.

**Table 5 healthcare-12-01346-t005:** Multivariable linear regression models of general health literacy and digital health literacy by socio-demographics and use of digital resources (standardized coefficients (β)).

Predictors	General Health Literacy (GEN-HL)	Digital Health Literacy (HL-DIGI)
Gender	−0.05	0.02
Age in years	0.19 *	0.02
Years in role	−0.12	−0.13
Teacher qualification	0.03	−0.07
Classes taught by teacher	−0.06	0.02
Use of digital resources		0.07

Note: Gen-HL/HL-DIGI score: from 0 = minimal health literacy to 100 = maximal health literacy. Gender: 0 = male, 1 = female. Years in role: from 1 = ≥10 to 4 = 30+. Teacher qualifications: 1 = teacher, 2 = senior teacher, 3 = teacher methodologist, 4 = expert. Classes taught by teacher: 1 = only primary, 2 = basic, 3 = secondary. * *p* < 0.05.

**Table 6 healthcare-12-01346-t006:** Health behavior and lifestyle and self-rated health.

Indicators	Frequency of Practice			
	Never	Occasional Use	Light Use	Heavy Use
Physical activity *n* (%)	2 (0.6)	3 (1.9)	90 (27.1)	234 (70.5)
Mean (SD) of days per week	4.57 (1.62)			
Frequency of smoking *n* (%)	No	Yes, once	Yes, occasionally	Yes, everyday
Tobacco last year	271 (81.6)	9 (2.7)	23 (6.9)	29 (8.7)
Tobacco last 30 days	280 (84.3)	7 (2.1)	18 (5.4)	27 (8.1)
E-cigarette last year	307 (92.5)	8 (2.4)	9 (2.7)	8 (2.4)
E-cigarette last 30 days	315 (94.9)	3 (0.9)	5 (1.5)	9 (2.7)
Frequency of alcohol use *n* (%)	No	Yes, once	Yes, occasionally	Yes, everyday
Alcohol last year	49 (14.8)	75 (22.6)	204 (61.4)	4 (1.2)
Alcohol last 30 days	80 (24.1)	84 (25.3)	163 (49.1)	5 (1.5)
Self-reported health categories *n* (%)	Very good	Good	Fair	Bad
	42 (12.7)	200 (60.2)	84 (25.3)	6 (1.8)
BMI categories *n* (%)	Normal weight	Overweight		Obese
	158 (47.6)	140 (42.2)		34 (10.2)
BMI score means (SD)	25.63 (3.51)			

Note: BMI: body mass index.

**Table 7 healthcare-12-01346-t007:** Pearson’s correlations between general health literacy and indicators of health behaviors and lifestyles.

	1	2	3	4	5	6	7	8	9
1. GEN-HL									
2. Physical activity	0.11 *								
3. Tobacco last year	−0.01	−0.07							
4. Tobacco last 30 days	0.01	−0.04	0.96 **						
5. E-cigarette last year	−0.04	−0.03	0.35 **	0.33 **					
6. E-cigarette last 30 days	0.03	−0.07	0.34 **	0.33 **	0.91 **				
7. Alcohol last year	−0.04	−0.15 **	0.21 **	0.17 **	0.12 *	0.10			
8. Alcohol last 30 days	0.01	−0.09	0.21 **	0.20 **	0.16 **	0.14 **	0.79 **		
9. BMI	−0.05	−0.11 *	0.12 *	0.11 *	0.04	0.04	0.16 **	0.14 **	
10. Self-reported health	0.16 **	0.19 **	−0.18 **	−0.15 **	−0.15 **	−0.15 **	−0.13 *	−0.13 *	−0.26 **

Note: GEN-HL = general health literacy. * *p* < 0.05; ** *p* < 0.01.

**Table 8 healthcare-12-01346-t008:** Multivariable linear regression models of health behavior indicators by general health literacy and socio-demographics (standardized coefficients (β)).

Predictors	Physical Activity	Tobacco Last Year	Tobacco Last 30 Days	e-Cigarette Last Year	e-Cigarette Last 30 Days	Alcohol Last Year	Alcohol Last 30 Days	BMI	Self-Perceived Health
Gender	0.04	−0.14 **	−0.13 *	−0.11 *	−0.16 **	−0.16 **	−0.17 **	−0.39 ***	0.08
Age in years	0.31 ***	0.03	−0.01	−0.11	−0.18 *	−0.04	−0.10	−0.11	0.14
Years in role	−0.31 **	−0.11	−0.10	−0.14	−0.11	0.01	0.17	0.27 ***	−0.34 ***
Qualification	0.02	−0.02	−0.03	0.06	0.07	0.05	−0.01	−0.16 **	0.14 *
Classes touched	0.20 ***	−0.02	−0.02	−0.05	−0.09	−0.02	−0.05	0.03	0.05
GEN-HL	0.09	−0.02	−0.01	−0.05	0.03	−0.05	−0.01	−0.05	0.15 **
R2	0.08	0.04	0.04	0.08	0.11	0.03	0.04	0.19	0.07

Note: General health literacy score: from 0 = minimal health literacy to 100 = maximal health literacy. Gender: 0 = male, 1 = female. Years in role: from 1 = ≥10 to 4 = 30+. Teacher qualifications: 1 = teacher, 2 = senior teacher, 3 = teacher methodologist, 4 = expert. Classes taught by teacher: 1 = only primary, 2 = basic, 3 = secondary. Physical activity: from 1 = never to 4 = heavy. Smoking and alcohol: from 1 = no to 4 = yes, every day. BMI: used as continuous variable. Self-perceived health: from 1 = very bad to 4 = very good. * *p* < 0.05, ** *p* < 0.01, *** *p* < 0.001.

## Data Availability

The data that support the findings of this study are available from the corresponding author, (S.S.), upon reasonable request.

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
