# Peer review of "Physical Education Teachers’ Health Literacy: First Evidence from Lithuania"

_healthcare, 2024, doi:10.3390/healthcare12131346_

Round 1
Reviewer 1 Report
Comments and Suggestions for Authors
Thank you very much for submitting this contribution for peer review. I am convinced that this work can make a substantial contribution to the existing literature. In reading this work, I do still have a few issues where I question and think they need clarification or modification.
There are still writing and grammatical errors here and there that need attention. Those that I detected I will mention.
This study deals with health literacy in the field of physical education. As you state at the outset of this work itself is that health literacy is a concept that is both new and old, I think it makes sense to frame that in the domain of physical education, it is increasingly common to talk about physical literacy and that health literacy there can be a component within the school subject of Physical Education. This needs attention to better interpret the place of ‘health literacy’ in PE. I think it also makes sense to recognise that another part (especially the knowledge-related elements) should be given a place elsewhere in the school curriculum and that a communication and alignment between the PE teacher and the Biology teacher, for example, can be crucial to give this topic a full place in the whole of pupils' HL development.
Somewhere in the work, it seems useful to clarify what the education of a PE teacher can or has contributed to personal health literacy.
On page 2, line 57, after reference 21 and up to reference 14: this sentence does not read smoothly, making it seem like it hangs in between.
The sentence from line 63 to 65, ending with references 28 and 29, lacks coherence and interrupts the flow of the text, leaving me as a reader unsure of how it fits into the overall context of the outlined issue.
The entire following paragraph needs more context to explain its relevance in the development of health literacy among students.
On page 3, line 105, I think it makes sense to use "related to school children's health education" instead of "related to education of school children's health."
The paragraph on page 3 that begins on line 112 and ends on line 115 presents a very narrow and outdated view of education. For several decades now, there has been a consensus within educational science and pedagogy that designing a powerful and efficient learning process cannot be a transfer of knowledge from teacher to student. A constructivist approach to the learning process appears to be a more optimal practice.
On page 3, line 120: use "A" instead of "the" study
line 56 a bracket is missing in front of reference 56.
On page 6, line 254, you use APRP for the first time Any abbreviation used for the first time should be written out in full In line 257, there is a reference in a different citation style (Pelican et al., 2022)
On page 7 in the results section, I read that the average age of the teachers who participated in the study is 51 years. Since this is a fairly mature audience, I think it would be useful to include this in your discussion and to discuss it critically in relation to the results of this study.
On page 7, table 1 last line of the table there is a disk error that causes what it says to have a very sensitive connotation: "Classes touched by teacher" seems to me to be: "Classes taught by teacher"?
On pages 8 and 10, figures 1 and 4 are not an appropriate representation to place results in a scientific publication. I think it would be useful to check here whether these can be integrated in a more correct scientific way.
On page 10, the difference between the outcomes to measure effect size is significantly different. The reason you give for doing both tests is because there is criticism of one of them. For that reason, I think it is important to bring this up in your discussion and discuss it critically so that the reader of your work understands what this difference means for interpreting the other results.
The layout of the tables needs to be adapted to the guidelines in several places. There seem to me to be too much intermediate line at the moment.
You use a rough estimate of the teacher's BMI to tell us something about the health status of the teachers studied. A critical remark about the use of BMI as a measure of health status in this specific population is that the morphology (and therefore also muscle mass) in this target group can be an explanatory factor for a higher BMI and therefore does not necessarily indicate an unhealthy lifestyle. This perspective is missing from this work.
Page 17 line 661 an s is missing from "insight" it should be plural I think "insights"
Comments on the Quality of English Language
I included my Language related suggestions in the text above.
A general remark; To increase readability you may consider to shorten some of the longer sentences in this work.
Reviewer 2 Report
Comments and Suggestions for Authors
The topic "Physical Education Teachers’ Health Literacy: First Evidence from Lithuania" is significant. It highlights the relationship between health literacy and a health-promoting lifestyle among physical education teachers in Lithuania, who, by virtue of their subject, should be role models for health-promoting behaviors among students. This is also important due to comparative studies in other countries (research conducted using the European Health Literacy Survey HLS19-Q12).
The presented research results confirm this. The research methodology is described very precisely, and the results are presented clearly and comprehensibly. The study poses a research question. Given that the correlation between variables was examined, it is worth considering the formulation of research hypotheses.
Technical notes: It is necessary to add the following under the tables and charts:
Source: write "Source: own elaboration" under the tables
Recommendation: The work can be accepted in its presented form or supplemented with formulated and verified research hypotheses.
Reviewer 3 Report
Comments and Suggestions for Authors
In recent years, researchers in all fields have become increasingly interested in the concept of literacy. There is a growing body of research on physical literacy, or health literacy as in this case.
In these concepts of literacy, different dimensions are brought together to form a single construct. Among these dimensions are status, level of knowledge and motivation. It is not only important to have a high level in one of the dimensions, but it is necessary to have a high level in all of them. Knowledge without motivation is not enough and so attempts are being made to improve the literacy outcome of the population in different domains, such as in the context of physical education.
The authors have done a great job in preparing their research and this manuscript.
I would have liked to find the Strobe checklist for cross-sectional studies. I would recommend that it be followed and included in the material and method. Indicate which items were completed and where in the manuscript.
One of the limitations I had found was included by the authors as a precautionary measure in the limitations of their study. It would have been necessary to carry out a reliability study, carrying out a test-retest. It would have been advisable. Future studies could give more consistency to the results.
Best regards
Reviewer 4 Report
Comments and Suggestions for Authors
The research presented by the authors is of great importance for the infant-youth population. The health threats to children are becoming more and more serious. Many times these threats are a consequence of bad habits and misinformation, so the role of teachers in general and Physical Education teachers in particular, is fundamental.
The introduction is pertinent to the objective of the research, exhaustive and gathers a large number of current documents to give a good empirical solidity.
In reference to the materials and methods, all the subsections are very complete, with great detail and in correspondence with the objective of the research. It is recommended that the questionnaires used in the research be included as complementary material in the document.
In the same line as in the previous sections, the results are presented in a clear and very detailed manner. The large number of tables and graphs makes it easier to read and interpret the results. Perhaps Figure 3 needs a larger size or better quality, as it is difficult to see some of the numbers correctly.
The discussion also presents a solid structure and current references. Once again, it is a very complete section that contributes great quality to the research.
In conclusion, I believe that this is a fundamental study to continue increasing the health of schoolchildren with properly trained health professionals, especially in line with the “One Health” concept. In addition, it should be noted that one of the pedagogical models in the field of physical education is precisely that of health, yet another argument for proper health training of teachers in this specialty.
Comments on the Quality of English LanguageMinor editing of English language required
Round 2
Reviewer 1 Report
Comments and Suggestions for Authors
Dear authors, my congratulations in advance on the way in which you have responded to the reviewers' suggestions, concerns and comments. You have managed to comply to a great extent. I still have one small concern that I would like to share with you. You collected your data via Google forms and therefore your collected raw research data is stored or has already been stored somewhere on a server outside your institution. My experience shows that this is not good practice in terms of GDPR. I think it would be advisable to discuss this with your ethics committee or data protection officer and see how you can better protect your raw research data against improper use. Sorry I didn't mention that in my previous review, but it slipped my mind.
Kind regards
Reviewer 3 Report
Comments and Suggestions for Authors
The authors included my requests in this round of review and I have no objection to recommending publication of this manuscript if the other reviewers and the editor agree.
Round 3
Reviewer 1 Report
Comments and Suggestions for Authors
All points raised have been addressed correctly.